# Effect of Flux Barriers on Short-Circuit Current and Braking Torque in Dual Three-Phase PM Machine

**Lin Liu, Kai Wang \*, Lingling Guo and Jian Li**

College of Automation, Nanjing University of Aeronautics and Astronautics, 29 Jiang Jun Road, Nanjing 211106, China; linliu@nuaa.edu.cn (L.L.); llingguo@nuaa.edu.cn (L.G.); jli@nuaa.edu.cn (J.L.)
\* Correspondence: k.wang@nuaa.edu.cn

**Abstract:** This paper investigates the influence of stator flux barriers on the short-circuit current (SCC) and braking torque of a dual three-phase permanent magnet (PM) synchronous machine. By optimizing the position and width of stator flux barriers, the machine has a lower amplitude of short-circuit current and brake torque when the short-circuit fault occurs. First, the SCC and braking torque are analytically derived. The amplitude of SCC is proportional to the PM flux linkage and inversely proportional to the inductance. The braking torque is proportional to the square of the PM flux linkage and inversely proportional to inductance. Then, the equivalent magnetic circuit model of flux barriers is established. Its influence on flux linkage and inductance is analyzed, and the improvement mechanism of output torque and fault tolerance is revealed. Furthermore, the flux barriers' width is optimized by finite element analysis and the theoretical analysis is verified. Finally, experiments on the prototype machine are carried out for the validation.

**Keywords:** permanent magnet machine; dual three-phase winding; stator flux barriers; short circuit current; braking torque



## 1. Introduction

Multi-phase machines have been widely used in aviation, aerospace, and electric vehicles due to their high torque density, low torque ripple, and high reliability [1–3]. As a specific multi-phase machine, dual three-phase machines have received extensive attentions in recent years [4–6]. A complete fault-tolerant system needs not only fault-tolerant control, but also the corresponding fault-tolerant machines. Higher fault-tolerance capability requires that the fault phase does not affect the healthy phase, and that it has a smaller short-circuit current (SCC) and braking torque [6–10].

It is well known that higher self-inductance can suppress the SCC and braking torque, and lower mutual inductance can reduce the influence of the fault phase relative to the healthy phase. In [6–8], the influence of slot/pole mating on inductance and output characteristics was studied. However, the SCC, braking torque, and fault tolerant operation were not analyzed. In [9,10], based on the particularity of two sets of three-phase windings of a dual three-phase machine, the influence of the deviation angle of two sets three-phase windings on the stator magneto motive force (MMF) harmonics, torque capability, torque ripple, and eddy-current losses of the machine were analyzed. Then in [11], a special 15° phase shift was proposed to suppress the SCC of a 24-slot/10-pole dual three-phase machine. However, in [12], compared with other phase shifts, the 15° phase shift had the largest SCC amplitude. This phenomenon is due to the fact that 1st-order stator MMF harmonic suppression effect is different [13]. In [13], for a 48-slot/22-pole dual three-phase machine, the phase shift of 30° and 7.5° had a smaller SCC. It is worth noting that the machines above all adopted the overlapping winding structure with a coil span greater than 1. The winding structure not only had a long end, but the two sets of three-phase winding also had an electrical and magnetic coupling. Therefore, concentrated windings

are more suitable for fault-tolerant machines. In [14–16], various concentrated winding configurations of 12-slot/10-pole dual three-phase windings were taken into account. The results showed that the single-layer (SL) concentrated winding had the maximum self-inductance and the minimum mutual inductance, which can improve the fault tolerance of the machine. However, the MMF harmonics of SL concentrated winding is greater than that of double-layer (DL) concentrated winding and distributed winding, which increases the losses, vibration, and noise of the machine. Therefore, one of the difficulties in designing a fault-tolerant PM machine is how to improve the fault-tolerant capacity of the machine while ensuring that the other electromagnetic characteristics are not weakened.

Furthermore, the stator flux barrier structure was proposed in [17–23]. This design not only effectively suppressed the sub-harmonics of the stator MMF, but also realized the electrical, magnetic, thermal, and physical isolation between phases, thus effectively improving the fault tolerance of the machine. Then, in [22,23], it was found that the stator flux barriers can effectively reduce the SCC of the machine. However, it is worth noting that the flux barriers studied in the above papers were all located on the auxiliary teeth, and this will severely reduce the torque density of the machine with the slot number greater than the pole number. The effects of different position of flux barriers on SCC, braking torque, and fault-tolerant operation are rarely investigated. Therefore, this paper mainly focuses on the evaluation and comparison of the influence of different stator flux barrier positions and width on the SCC and braking torque for the 12-slot/10-pole dual three-phase machine, and carries out deep analysis of fault-tolerant operation.

The purpose of this paper is to reveal the principle of SCC and braking torque suppression of a dual three-phase machine with stator flux barrier structure, so as to improve the fault-tolerant operation ability of the machine. In Section 2, different structures of stator flux barriers are proposed. Then, the analytical expressions of SCC and braking torque for single-phase short-circuit fault are derived, and the effect of the stator flux barriers on flux linkage and inductance is revealed by simplified equivalent magnetic circuit model in Section 3. In Section 4, the 12-slot/10-pole PM machine with different stator flux barriers is designed, and the electromagnetic and short-circuit performances are comprehensively calculated by using the finite-element method (FEM) to verify the theoretical analysis. In Section 5, experiments on the prototype machine are carried out for validation. Finally, conclusions are drawn in Section 6.

## 2. Machine with Different Flux Barriers and Winding Distribution

In this section, the cross-section models of a traditional stator machine and stator flux barrier machine are first presented, as shown in Figure 1. The main parameters of the machine are shown in Table 1. There are two kinds of winding structures for the traditional 12-slot/10-pole dual three-phase machine, which are single-layer (SL) concentrated winding and double-layer (DL) concentrated winding respectively. Figure 1a shows the machine with the traditional stator with DL winding (DL-TSM), and two sets of three-phase winding deviations of 30°. Figure 1b shows the machine with the traditional stator with SL winding (SL-TSM), and two sets of three-phase winding deviations of 60°. Compared with the traditional dual three-phase machine, the stator of a flux barrier machine is composed of six E-cores. There is a flux barrier between each E-core and the adjacent core. Two types of flux barriers are used in the stator: the armature tooth flux barriers and the auxiliary tooth flux barriers. Figure 1c shows the stator with armature tooth flux barriers and SL windings. Figure 1d shows the machine with auxiliary tooth flux barriers and SL windings.

The above four kinds of machines adopt the same rotor, and the permanent magnet material is NdFeB 35SH. The DL winding structure has two coils for each phase, and the SL winding structure has one coil for each phase. However, to highlight the effect of stator flux barriers, the number of turns per phase is the same for all investigated machines.

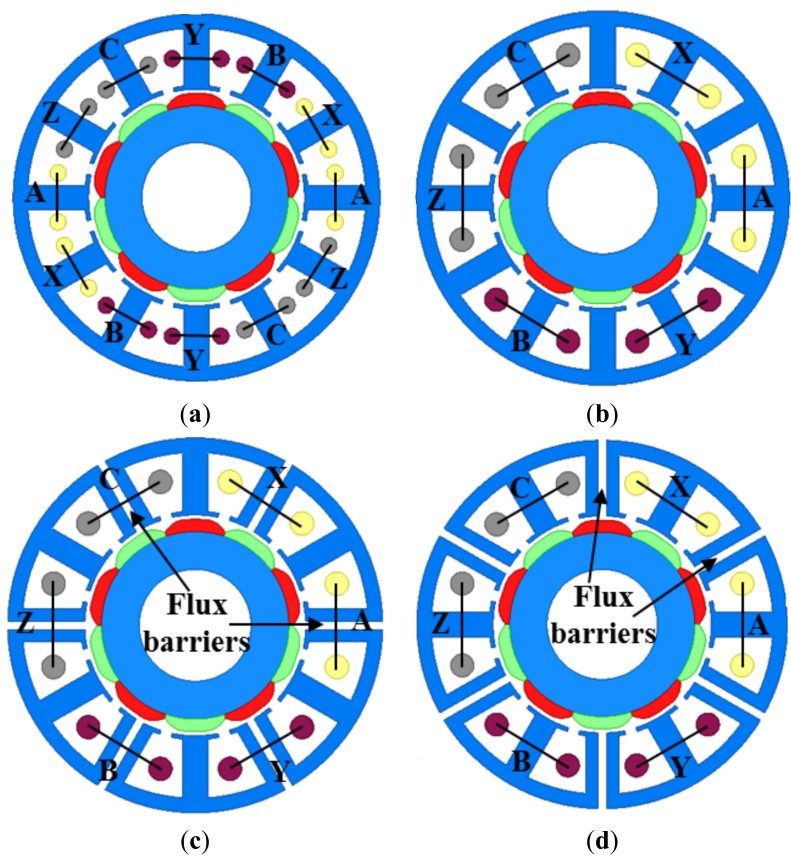

**Figure 1.** Cross sections of 12-slot/10-pole dual three-phase machines. (**a**) DL-TSM. (**b**) SL-TSM. (**c**) SL-ATM. (**d**) SL-ABM.

**Table 1.** Main Machine Parameters.

| Parameter | Value | Parameter | Value |
|---|---|---|---|
| Stator slot number | 12 | Stack length (mm) | 50 |
| Rotor pole number | 10 | Air-gap length (mm) | 1 |
| Rated speed (r/min) | 1000 | Rotor outer radius (mm) | 25 |
| Rated torque (N·m) | 4 | Rotor inner radius (mm) | 15 |
| Stator outer radius (mm) | 45 | Magnet thickness (mm) | 3 |
| Stator inner radius (mm) | 26 | Flux barrier width (mm) | 2.5 |
| Tooth width (mm) | 6 | NdFeB 35 remanence (T) | 1.2 |
| Yoke thickness (mm) | 3 | Number of turns/phase | 80 |

## 3. Influence of Stator Flux Barriers on SCC and Braking Torque

### 3.1. SCC and Braking Torque Calculation

The winding short circuit fault is one of the common fault types in the machine. Short circuit positions can be divided into winding end short-circuits and inter-turn short-circuits. Compared with inter-turn short-circuit, the possibility of short-circuit at winding ends is less. Usually, in order to alleviate the inter-turn short-circuit fault, the winding ends are artificially short-circuited [24]. The equivalent circuit of the healthy condition of A-phase winding and the short-circuit at the end is shown in Figure 2.

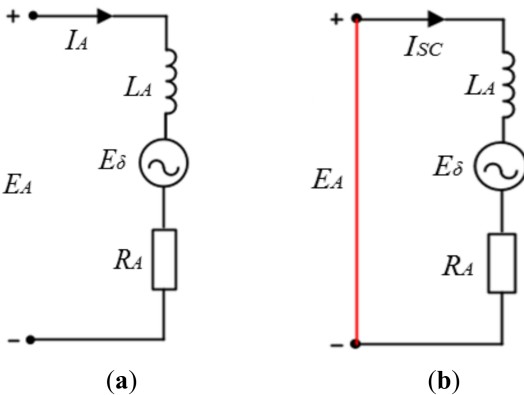

**Figure 2.** Equivalent circuit. (**a**) Healthy winding. (**b**) Single phase short circuit fault.

When a short circuit occurs at the winding end, the terminal voltage is 0, and the voltage equation when a fault occurs is shown in Equation (1).

$$E_A = L_A \frac{di_{SCC}}{dt} + E_\delta + R_A i_{SCC} = 0 \tag{1}$$

where $L_A$ is the inductance of phase A, $R_A$ is the resistance of phase A, $i_{SCC}$ is the short-circuit current, $E_\delta$ is the induced voltage, $t$ is the time. The expression of SCC can be obtained to solve the above differential by using Laplace transform. The SCC can be expressed as [25,26]

$$i_{SCC}(t) = N\psi_m \omega \frac{R_A \sin(\theta_0 + \omega t) - \omega L_A \cos(\omega t)}{R_A^2 + (\omega L_A)^2} + \left[ i_A \cos(\theta_0 + \delta) + \frac{N\psi_m \omega (L_A \omega \cos\theta_0 - R_A \sin\theta_0)}{L_A^2 \omega^2 + R_A^2} \right] e^{-\frac{R_A}{L_A} t} \tag{2}$$

where $N$ is the number of turns per phase, $\psi_m$ is the PM flux linkage, $\omega$ is electric angular frequency, and $\theta$ is the electrical angle between the axis of the short-circuited coil and d-axis. $\theta_0$ is the initial value of $\theta$, $\theta = \theta_0 + \omega t$, the electrical angle $\delta = \arctan(-L_A \omega / R_A)$. Therefore, the steady-state SCC can be expressed as

$$i_{SCC\_steady}(t) = \frac{N\psi_m \omega \cos(\omega t + \theta_0 + \delta)}{\sqrt{L_A^2 \omega^2 + R_A^2}} \tag{3}$$

According to (3), $\omega L_A$ is much greater than $R_A$ ($\omega L_A \gg R_A$) for the machine, so the amplitude of steady SCC is given by

$$i_{SCC\_steady} = \frac{N\psi_m}{L_A} \tag{4}$$

From Equation (4), the amplitude of steady SCC is proportional to PM flux linkage and inversely proportional to inductance. When a short circuit fault occurs in the A phase, the output torque of the machine can be represented as [27]

$$
\begin{aligned}
T_{torque} &= \frac{i_{SCC} E_A + i_B E_B + i_C E_C + i_X E_X + i_Y E_Y + i_Z E_Z}{\omega} \\
&= \frac{\left( i_A \cos(\theta_0 + \delta) - \omega N\psi_m \frac{R_A \sin\theta_0 - \omega L_A \cos\theta_0}{R_A^2 + \omega L_A^2} \right) e^{-\frac{R_A}{L_A} t} \sin(\omega t + \theta_0)}{\omega} \\
&\quad + N\psi_m \frac{R_A \sin(\omega t + \theta_0) - \omega L_A \cos(\omega t + \theta_0)}{R_A^2 + \omega L_A^2} \sin(\omega t + \theta_0) + \frac{i_B E_B + i_C E_C + i_X E_X + i_Y E_Y + i_Z E_Z}{\omega}
\end{aligned}
\tag{5}
$$

From (5), the steady-state braking torque generated by the SCC is

$$T_{Braking\ torque}(t) = -\frac{(N\psi_m \omega)^2 \cos(\omega t + \theta_0) \sin(\omega t + \theta_0)}{\sqrt{L_A^2 \omega^2 + R_A^2}} \tag{6}$$

It can be seen from (6) that braking torque is not only related to self-inductance and flux linkage, but also to the speed and resistance. When the machine runs in the high-speed area, the $\omega L_A$ is much greater than $R_A$ ($\omega L_A \gg R_A$) for the machine, so the amplitude of steady braking torque is given by [26]

$$T_{\text{Braking torque}} = -\frac{\omega N^2 \psi_m^2 \sin(2\theta)}{2L_A} \tag{7}$$

Obviously, the amplitude of braking torque is proportional to the square of PM linkage and inversely proportional to the inductance. However, it is worth noting that when a short-circuit fault occurs, it brings a significant torque ripple of twice the electrical frequency. Two-phase short circuits and three-phase symmetric short circuits can be analyzed similarly and will not be described here. Overall, both short-circuit current and braking torque are ultimately related to the flux linkage and inductance. The fault-tolerant capacity of machine can be improved effectively by optimizing flux linkage and inductance.

### 3.2. Magnetic Flux Analysis with Flux Barriers

According to the above analysis, the permanent magnet flux linkage is related to the SCC amplitude and the maximum braking torque. Therefore, it is necessary to accurately calculate the influence of different stator structures on the PM flux linkage. For the dual three phase permanent magnet machine, the PM flux linkage can be expressed as

$$\psi_m = k_w N R_o L_{ef} \int B_g(\alpha, t) d\alpha \tag{8}$$

where $k_w$ is the winding coefficient, $B_g$ is the no-load air-gap flux density, $\alpha$ is the mechanical angle, $R_o$ is the outer radius of rotor, and $L_{ef}$ is the lamination length. For a 12-slot/10-pole dual three-phase machine, the distribution coefficients of SL winding structure and DL winding structure are 1, so the winding coefficient is determined by the short distance coefficient. However, the existence of stator flux barriers changes the machine short-distance coefficient, and different positions and widths have different influences on the short-distance coefficient, as shown in Table 2 and Figure 3.

**Table 2.** Winding Coefficient for Dual Three-Phase Machine with Different Winding Structure and Stator Structure.

| Machine Type | $k_d$ | $k_p$ | $k_w = k_d \cdot k_p$ |
|---|---|---|---|
| DL-TSM | 1 | $\sin\left(\frac{\tau_s}{\tau_p} \cdot \frac{\pi}{2}\right)$ | 0.966 |
| SL-TSM | 1 | $\sin\left(\frac{\tau_s}{\tau_p} \cdot \frac{\pi}{2}\right)$ | 0.966 |
| SL-ATM | 1 | $\sin\left(\frac{(\tau_s+\Delta)}{\tau_p} \cdot \frac{\pi}{2}\right)$ | 1 |
| SL-ABM | 1 | $\sin\left(\frac{(\tau_s-\Delta)}{\tau_p} \cdot \frac{\pi}{2}\right)$ | 0.928 |

Where $\tau_s = 2\pi/N_s$ is the slot pitch, $\tau_p = 2\pi/(2p)$ is the pole pitch, $\Delta$ is the radius of the stator flux barrier, $\beta$ is the width of the stator magnetic barrier, and $R_i$ is the stator inner diameter, $\Delta = \text{asin}(\beta/2R_i)$. Table 2 shows the winding coefficients of the machines when the width of the stator flux barrier is 2.5 mm. Figure 3 shows the variation of the winding coefficient of the stator flux barrier machine with the width of the flux barriers.

The no-load air-gap flux density can be calculated by the following expression:

$$B_g(\alpha, t) = F(\alpha, t) \times \Lambda(\alpha) \tag{9}$$

where $F$ is PM MMF and $\Lambda$ is the air-gap permeance. Figure 4 shows the stator flux line paths of different machines. It can be seen that the presence of stator flux barriers changes the flux line paths.

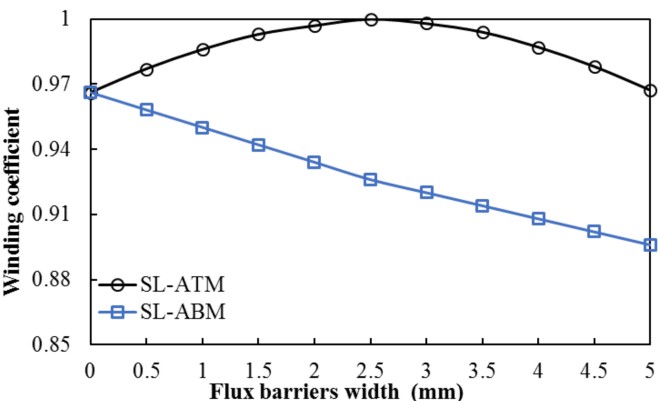

**Figure 3.** Winding coefficient versus flux barrier width for machines.

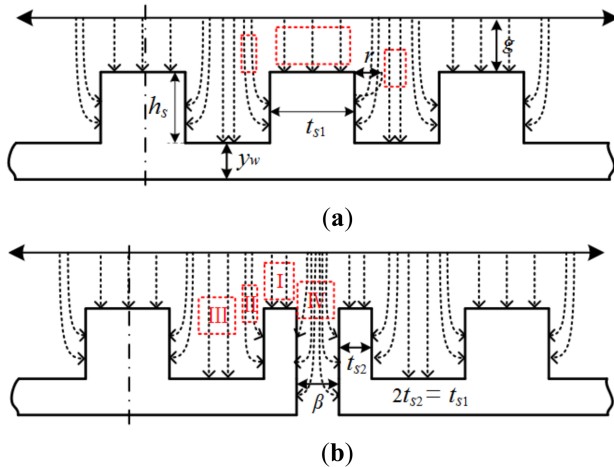

**Figure 4.** Path of flux lines in air-gap for different machines. (**a**) Traditional stator machine. (**b**) The machine with stator flux barriers.

According to the path of flux lines shown in Figure 4, when the flux lines go directly to the stator tooth (part I), the air-gap permeability can be expressed as

$$\Lambda_1 = \frac{\mu_0}{g + (\mu_r/h_m)} \tag{10}$$

When the flux lines go through the slot (part II and part III), $\Lambda$ can be expressed as

$$\Lambda_{2,3} = \begin{cases} \frac{2\mu_0}{2(g+(\mu_r/h_m))g+\pi r_1}, 0 < r_1 < \frac{2h_s}{\pi} \\ \frac{\mu_0}{g+(\mu_r/h_m)+h_s}, r_1 \geq \frac{2h_s}{\pi} \end{cases} \tag{11}$$

The stator barrier can be thought of as a slot; the slot depth is infinite, and the width of the slot is $\beta$. When the flux lines go through stator flux barriers (part IV), $\Lambda$ can be expressed as

$$\Lambda_4 = \begin{cases} \frac{2\mu_0}{2(g+(\mu_r/h_m))g+\pi r_2}, \beta \leq r_2 \\ 0, \beta > r_2 \end{cases} \tag{12}$$

where $\mu_0$ is the air-gap permeability, $\mu_r$ is the PM permeability, $g$ is the air-gap length, $h_m$ is the magnet thickness, $h_s$ is the slot depth, $\beta$ is the width of flux barriers, and $r_1$ and $r_2$ are

the flux path lengths in slot openings and flux barriers. In addition, the air-gap permeance $\Lambda(\alpha)$ due to the slotting can be given by ignoring the high-order harmonics, i.e., [26]

$$\Lambda(\alpha) = \sum_{i=0,1,2...}^{\infty} \Lambda_i \cos\left(i\frac{N_s}{2}\alpha\right) \tag{13}$$

where $N_s$ is the stator slot number and $\Lambda_i$ is the amplitude of $i$th order harmonic of air-gap permeance. The rotors of the four different machines are consistent, so the rotor magnetomotive force can be expressed as

$$F(\alpha,t) = \sum_{j=1,3,5...}^{\infty} \frac{gh_m + \mu_r}{\mu_0 h_m} B_r \sin(jp_r\alpha - j\omega t) \tag{14}$$

where $B_r$ is the PM remanence and $p_r$ is the pole pair of the machine.

By substituting (13) and (14) into (8), the PM flux linkage can be expressed as

$$\psi_m = \sum_{i=0,1,2...}^{\infty} \sum_{j=1,3,5...}^{\infty} \frac{F_j\Lambda_i Nk_w L_{ef} R_o}{i\frac{N_s}{2} \pm jP_r} \times \sin\left[\left(i\frac{N_s}{2} \pm jp_r\right)\frac{2\pi}{N_s}\right] \cos(\pm jp_r\omega t) \tag{15}$$

An analytical model similar to the above air-gap permeance can be used regardless of the presence of flux barriers on the stator, and the calculation method of the PM flux linkage is basically the same, so it will not be described again. There is a slight deviation between the analytical results and the finite element analysis (FEA) because the saturation of the stator core is ignored in the analytical process. Figure 5 shows the variation of the PM flux linkage of different machines with the width of the flux barriers, which the trend of analytical calculation is consistent with that of FEA. It can be seen that the PM flux linkage of the SL-ATM will first increase and then decrease with the increase of the flux barrier width. However, the PM flux linkage of the SL-ABM will decrease with the increase of the flux barrier width.

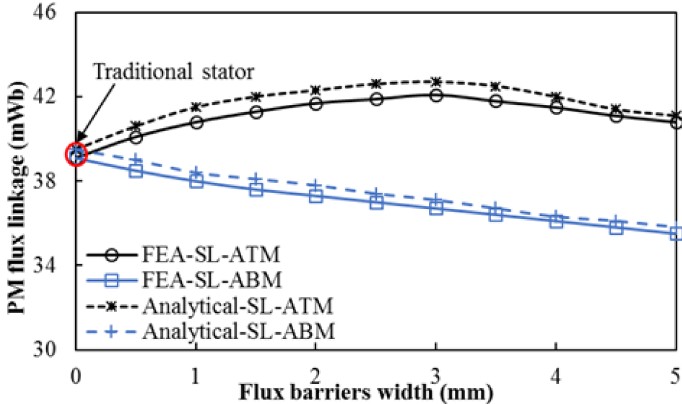

**Figure 5.** Variation of the PM flux linkage of different machines with the width of the flux barriers.

### 3.3. Inductance Analysis with Flux Barriers

As mentioned above, a larger self-induction can effectively suppress the short-circuit current and prevent damage to the machine windings. Due to the fact that the machine adopts a surface-mounted permanent magnet, the effective air gap of the machine is large, and the stator slot thickness of the permanent magnet fault-tolerant machine is high; the machine is not easily saturated, so the influence of the saturation of the ferromagnetic material of the machine can be ignored when solving the inductance analytical formula. According to the definition, the calculation formula of inductance is expressed as [28]:

$$L_A = \frac{\psi}{I} = \frac{N\Phi}{I} = \frac{NF_m}{IR_m} = \frac{N^2}{R_m} \tag{16}$$

where $I$ is the excitation current of the winding, $\psi$ is the magnetic flux linkage by the winding current, $\Phi$ is the magnetic flux, $N$ is coil turns per phase, $R_m$ is the total reluctance of the magnetic circuit, and $F_m$ is the stator magnetomotive force. According to (16), when the number of coil turns is fixed, the inductance is related to the reluctance of the machine and its distribution. According to the distribution of magnetic flux, the magnetic circuit model of one branch of A-phase winding of the armature tooth flux barrier machine is established, as shown in Figure 6. The magnetic circuit model includes the iron reluctance, the air-gap reluctance, the slot opening reluctance, the inter-slot reluctance, and the flux barrier reluctance. The pressure drop of magnetic flux on the iron material is ignored, so the magnetic circuit model can be simplified as shown in Figure 6c.

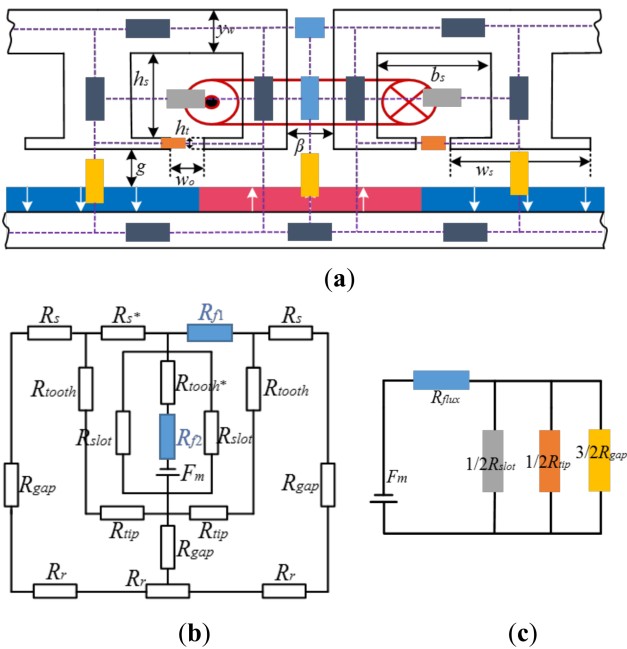

**Figure 6.** The equivalent reluctance network model of armature tooth flux barrier machine. (**a**) The cross section model of phase A. (**b**) Equivalent reluctance network model of phase A. (**c**) Simplified reluctance network model of phase A.

Where $R_{flux}$ is the reluctance of flux barriers, $R_{gap}$ is the air-gap reluctance, and $R_{slot}$ and $R_{tip}$ are the reluctance of inter-slot and slot opening. $R_{flux}$, $R_{slot}$, $R_{tip}$, and $R_{gap}$ can be expressed as

$$\begin{cases} R_{flux} = \dfrac{\beta}{\mu_0(h_s+h_t+y_w)L_{ef}}, R_{slot} = \dfrac{b_s}{\mu_0 h_s L_{ef}} \\ R_{tip} = \dfrac{w_o}{\mu_0 h_t L_{ef}}, R_{gap} = \dfrac{gh_m+\mu_r}{\mu_0 w_s h_m L_{ef}} \end{cases} \tag{17}$$

By substituting (17) into (16), the A-phase inductance of the armature tooth flux barrier machine can be obtained. Similarly, the A-phase inductance of DL winding with traditional stator machine, SL Winding with traditional stator machine, and auxiliary tooth flux barrier machine can be obtained by the method. Table 3 shows A-phase self-induction of the four machines.

**Table 3.** Self-inductance for Different machines.

| Machine Type | $L_A$ (mH) | Machine Type | $L_A$ (mH) |
|---|---|---|---|
| DL-TSM | 1.08 | SL-TSM | 2.14 |
| SL-ATM | 2.19 | SL-ABM | 2.12 |

However, when the width of the flux barrier increases, the slot width decreases and the leakage inductance in the slot increases. Meanwhile, the width of the tooth tip also

decreases with the increase of the width of the flux barrier, and the slot opening leakage inductance will also increase. Figure 7 shows the variation of A-phase self-inductance with the flux barrier width of the machine.

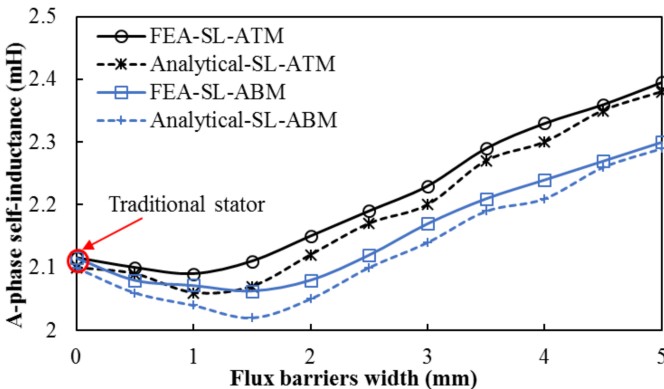

**Figure 7.** Variation of the A-phase self-inductance with the width of the flux barriers.

It can be seen from Figure 7 that the A-phase self-inductance of both SL-ATM and SL-ABM decreases first and then increases with the increase of the width of the flux barriers. When the decrease of air-gap inductance is greater than the increase of slot leakage inductance and tooth tip leakage inductance, the overall inductance of the machine shows a downward trend. When the decrease of air-gap inductance is less than the increase of slot leakage inductance and tooth tip leakage inductance, the overall inductance of the machine shows an upward trend.

## 4. Analysis and Comparison

To further explain the effect of stator flux barriers on machine fault-tolerant performance, four 12-slot/10-pole machines with different flux barriers and winding distribution were designed, as shown in Figure 1 and the main parameters are shown in Table 1. In this section, the electromagnetic characteristics of four kinds of machines under short circuit conditions are compared.

### 4.1. Selection of Flux Barrier Width

An analytical model and FEA were used to analyze the variation of the peak SCC and the steady SCC with the width of flux barriers of the machines, as shown in Figure 8. It can be seen that the peak SCC and the steady SCC have the same trend with the change of the flux barrier width. The peak SCC and steady SCC of SL-ATM will increase first and then decrease with the increase of the magnetic barrier width. This is because when the width of the flux barrier is less than 1.5 mm ($0 < \beta < 1.5$), the PM flux linkage is increasing, while the self-induction is decreasing, so the SCC is increasing. When the width of the flux barrier is greater than 1.5 mm ($1.5 < \beta$), although the PM flux linkage of the machine is still increasing, the self-inductance of the machine is increasing more, so the SCC shows a decreasing trend. The peak SCC and steady SCC of the SL-ABM decrease with the increase of the width of the flux barriers. Figure 9 shows the variation of the braking torque with the width of flux barriers. Compared with the SL-TSM, the braking torque of SL-ATM first increases and then decreases with the increase of the flux barrier width, while the braking torque of SL-ABM decreases with the increase of the flux barrier width. After comprehensive consideration, we determined that the width of the stator flux barrier is 2.5 mm.

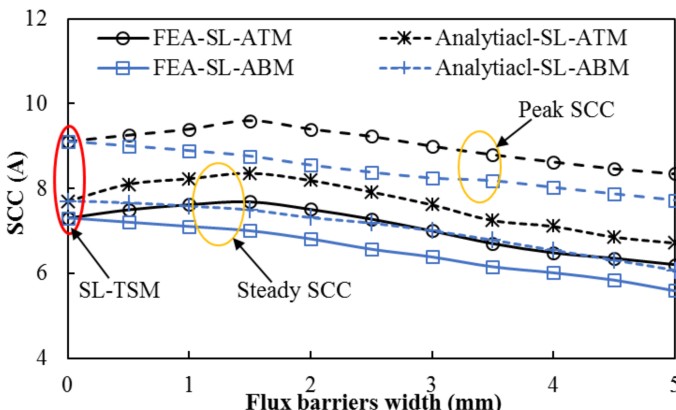

**Figure 8.** Variation of the peak SCC and the steady SCC with the width of flux barriers at 1000 r/min.

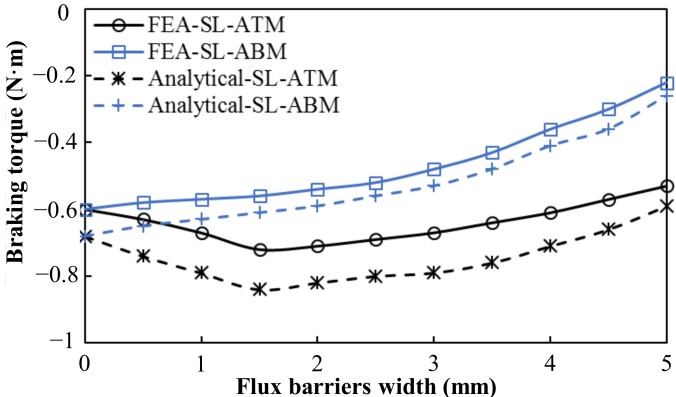

**Figure 9.** Variation of the braking torque with the width of flux barriers at 1000 r/min.

*4.2. Single-Phase Short Circuit Condition*

It is known that permanent magnet fault-tolerant machines must achieve magnetic isolation, physical isolation, and thermal isolation between phases, and effectively suppress the SCC. Therefore, the SL winding structure is very necessary. Figure 10 shows the single-phase SCC waveform of the four machines when the phase current is 5A, the speed is 1000 r/min, and the single-phase short-circuit suddenly occurs. Compared with SL winding, DL winding has smaller self-inductance, larger mutual-inductance, and the worst ability to inhibit the SCC. Therefore, the SCC of DL-TSM is the largest. Compared with the SL-TSM, when the width of the flux barriers is 2.5 mm, the self-inductances of SL-ATM and SL-ABM are larger than those without the flux barriers. Although the flux barriers make the PM flux linkage of SL-ATM increase, the SCC is still smaller than SL-TSM. On the contrary, the existence of flux barriers in SL-ABM leads to the decrease of PM flux linkage, so the SCC is inhibited.

Figure 11 shows the variation of the braking torque with speed when single-phase short circuit occurs. It can be seen that the braking torque of all four machines increases first and then decreases with the increase of speed, and there is maximum braking torque at the low speed. The braking torque of DL-TSM is the largest, because the self-inductance of DL winding structure is the smallest. Among SL winding machines, the SL-ATM has the largest braking torque, because the braking torque is not only inversely proportional to the self-inductance, but also proportional to the square of the PM flux linkage.

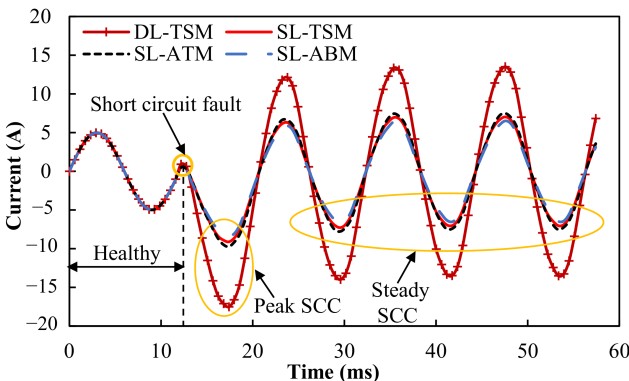

**Figure 10.** Single-phase SCC waveform of different machines at 1000 r/min.

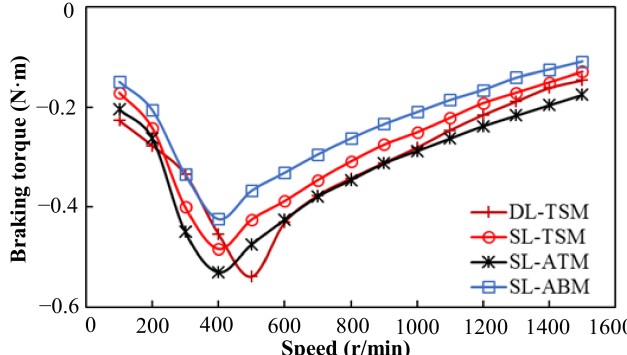

**Figure 11.** Variation of the braking torque with speed under single-phase short circuit condition.

Figure 12 shows the output torque after the single-phase short circuit fault of different machines. It can be seen that the average output torque of DL-TSM and SL-TSM under healthy conditions is 3.7 N·m, and the torque ripples are 2.2% and 2.8%, respectively. The output torque of the SL-ATM is 4.08 N·m, an increase of 10.27% compared with the traditional machine. However, due to the flux barriers, the 6th-order harmonic component is introduced into the cogging torque, which increases the torque ripple. Therefore, the torque ripple of the SL-ATM is 4.6%. The output torque of the SL-ABM is 3.3 N·m, and the torque ripple is 4.52%. When a single-phase short-circuit fault occurs, not only is the output torque of the machine slightly reduced, but the torque ripple also becomes larger. Taking the SL-ATM as an example, the output torque of the machine is reduced by 26.3%, and the torque ripple is increased to 137.6%. This is because the short circuit phase does not provide output torque, but produces a braking torque, which reduces the total output torque. For the asymmetric short circuit, the fault winding will generate negative sequence current. The magnetic field generated by the negative sequence current is synchronized with the rotor speed, but rotates in the opposite direction, so an even number harmonic of the output torque will be generated after the fault, which increases the torque ripple.

Table 4 is the comparison of electromagnetic performance of four kinds of machines under rated working conditions, in case of health and single-phase short-circuit fault.

**Table 4.** Electromagnetic performance under single-phase short circuit condition.

| Machine Type | Output Torque (N·m) | Torque Ripple (%) | Peak SCC (A) | Steady SCC (A) | Braking Torque (N·m) |
|---|---|---|---|---|---|
| DL-TSM | 3.7/2.6 | 2.2/113.3 | 18.3 | 14.6 | −0.3 |
| SL-TSM | 3.7/2.8 | 2.8/121.6 | 9.4 | 7.9 | −0.26 |
| SL-ATM | 4.1/3.2 | 4.6/137.6 | 10.2 | 8.5 | −0.31 |
| SL-ABM | 3.3/2.4 | 4.52/122.2 | 8.3 | 7.2 | −0.22 |

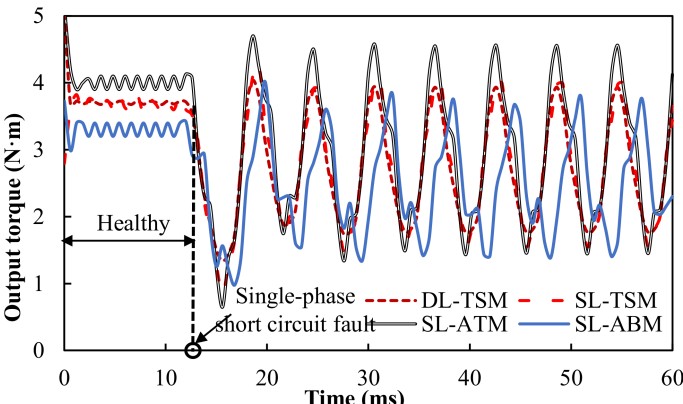

**Figure 12.** Output torque waveform of the machines when other healthy phases work normally after single-phase short circuit occurs at 1000 r/min.

### 4.3. Three-Phase Short Circuit Condition

According to the above analysis, the asymmetric short circuit in the winding will produce a large torque ripple. Therefore, for a dual three-phase PM machine, the windings that have a single-phase short-circuit fault can be artificially three-phase symmetrical short-circuited. The three-phase symmetrical short circuit can balance the asymmetry of the magnetic potential generated by the asymmetry of the short-circuit current, thereby reducing the torque ripple generated by the machine short-circuit fault. Then, through another set of healthy three-phase winding overload operations, the machine can reach the rated output torque requirements. Moreover, compared with the single-phase asymmetric short circuit, the active three-phase symmetric short circuit can suppress the SCC. Figure 13 shows the variation of the three-phase SCC of different machines with speed. It can be seen that both the peak SCC and the steady SCC in the three-phase short-circuit are reduced compared with the single-phase short-circuit. Meanwhile, the three-phase SCC of the DL-TSM is still the largest, and the three-phase SCC of the SL-ABM is still the smallest. The variation of the three-phase SCC with the width of the flux barrier is consistent with that of the single-phase SCC, so this paper will not repeat it.

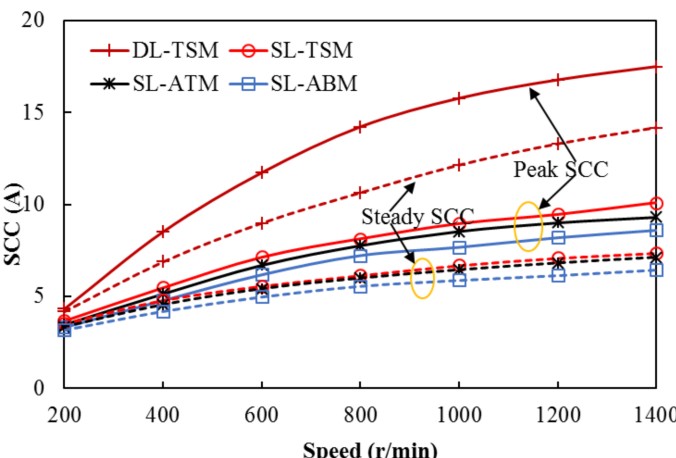

**Figure 13.** Variation of the three-phase SCC of different machines with speed.

Figure 14 shows the variation of braking torque with speed when three-phase short circuit occurs. Compared with the single-phase short circuit, the braking torque after a three-phase short circuit has a greater increase. However, its variation trend is consistent with a single-phase short-circuit fault, so it will not be described here.

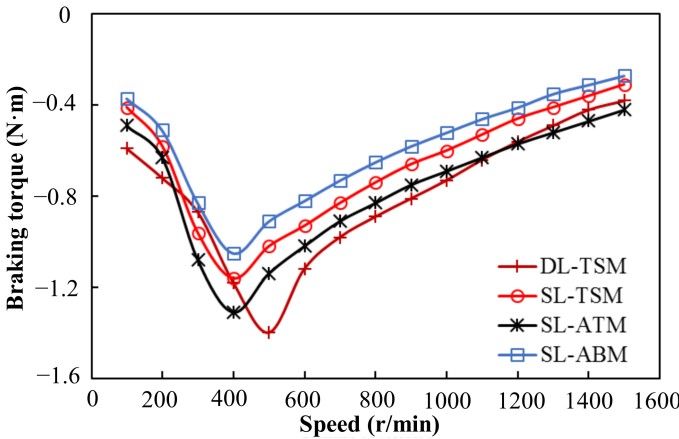

**Figure 14.** Variation of the braking torque with speed under three-phase short circuit condition.

The output torque of the machines when other healthy phases work normally after three-phase short circuit is shown in Figure 15. Compared with single-phase short circuits, three-phase short circuits can effectively restrain the torque ripple. Among them, the torque ripple of the DL-TSM is 24.6%, the SL-TSM is 36.5%, the SL-ATM is 39.2%, and the SL-ABM is 41.5%. However, the three-phase short circuit fault further reduces the output torque of the machine. At this time, the output torque of SL-ATM is the greatest, at 1.23 N·m. Therefore, although the braking torque of SL-ATM is the largest, the compensation current required for fault-tolerant operation at rated load is the smallest.

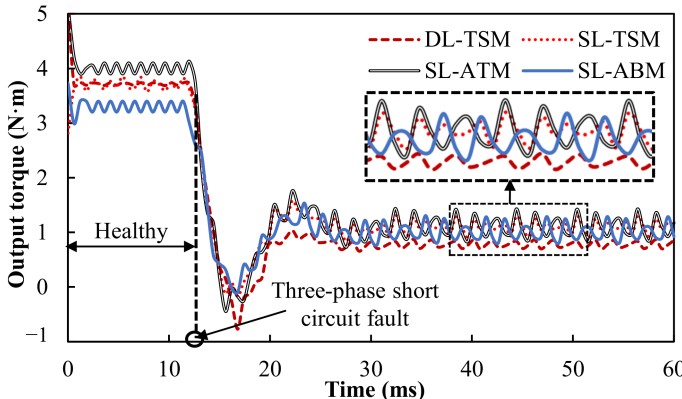

**Figure 15.** Output torque waveform of the machines when other healthy phases work normally after three-phase short circuit occurs at 1000 r/min.

Table 5 shows a comparison of the electromagnetic performance of four kinds of machines under rated working conditions, both healthy, and with a three-phase short-circuit fault.

**Table 5.** Electromagnetic performance under three-phase short circuit condition.

| Machine Type | Output Torque (N·m) | Torque Ripple (%) | Peak SCC (A) | Steady SCC (A) | Braking Torque (N·m) |
|---|---|---|---|---|---|
| DL-TSM | 3.7/0.83 | 2.2/24.6 | 18.3 | 14.6 | −0.82 |
| SL-TSM | 3.7/1.1 | 2.8/36.5 | 9.4 | 7.9 | −0.64 |
| SL-ATM | 4.1/1.2 | 4.6/39.2 | 10.2 | 8.5 | −0.77 |
| SL-ABM | 3.3/0.95 | 4.52/41.5 | 8.3 | 7.2 | −0.52 |

## 5. Experimental Validation

Through the above research, an experimental prototype machine was established, as shown in Figure 16. The specifications and main geometric parameters of the machine are given in Table 1. Figure 16a shows the stator lamination, which is composed of six E-type cores. The stator lamination cannot be fixed by jacket due to the air gap flux barriers between the E-type cores. Therefore, the stator and jacket adopt dovetail groove type overmatch, as shown in Figure 16b. The rotor stack and PMs are shown in Figure 16c. The experimental apparatus consists of dual three-phase inverters with a common DC power supply, and the hardware platform based on dspace-1007 is shown in Figure 16d. The experimental platform of the machine is shown in Figure 16e.

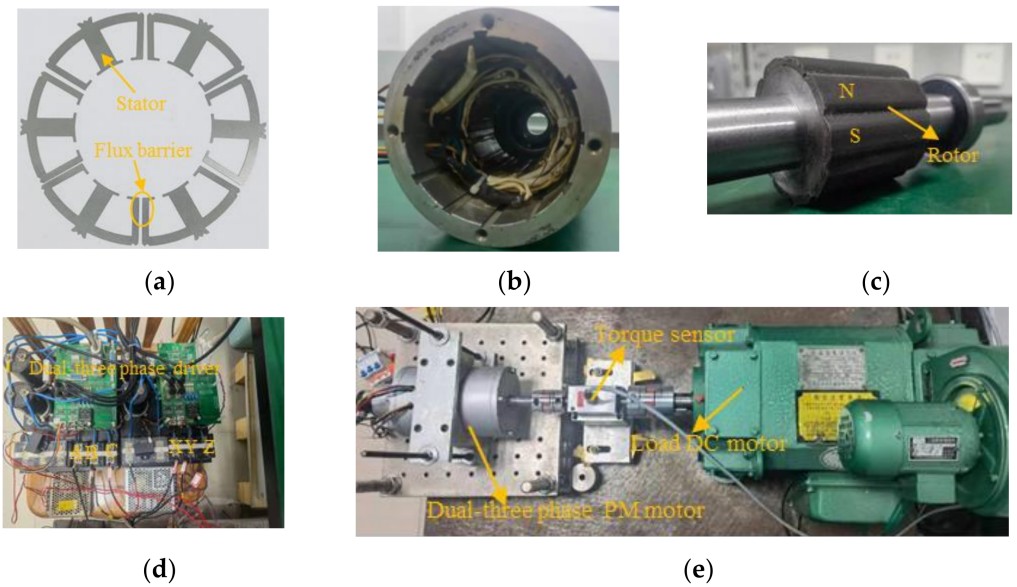

**Figure 16.** Machine prototype and test rig. (**a**) Stator lamination, (**b**) machine jacket, (**c**) rotor stack with magnets, (**d**) dual three-phase inverter, (**e**) test rig.

Figure 17 shows that the back-EMF with no-load at the speed is 1000 r/min; the measured results are in good agreement with the simulation results. Figure 18 shows the waveform of output torque and A-phase, X-phase current of dual three-phase windings working simultaneously in healthy condition. Figure 19a illustrates the output torque and SCC in the single-phase short-circuit fault. It can be seen that the torque ripple becomes larger and the short-circuit current is 7.5A, which is basically consistent with the simulation. Finally, the output torque waveform and SCC waveform under a three-phase short-circuit fault are shown in Figure 19b. Compared with the single-phase short circuit fault, not only is the torque ripple inhibited, but the SCC also changes from 7.5 A to 6.8 A.

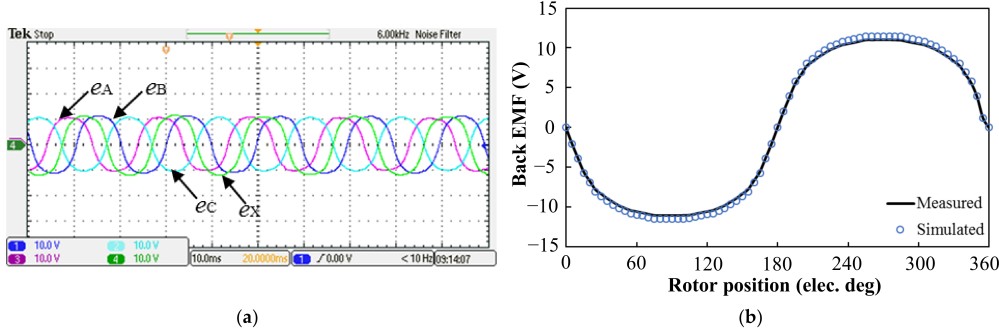

**Figure 17.** Back-EMF waveform with no-load at the speed is 1000 r/min. (**a**) Measured results, (**b**) simulated results.

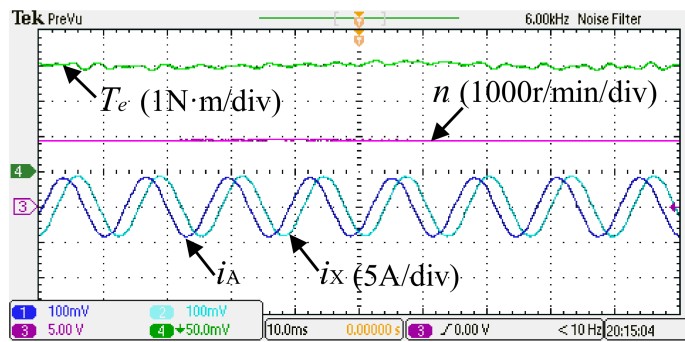

**Figure 18.** Output torque and phase current waveform under healthy condition.

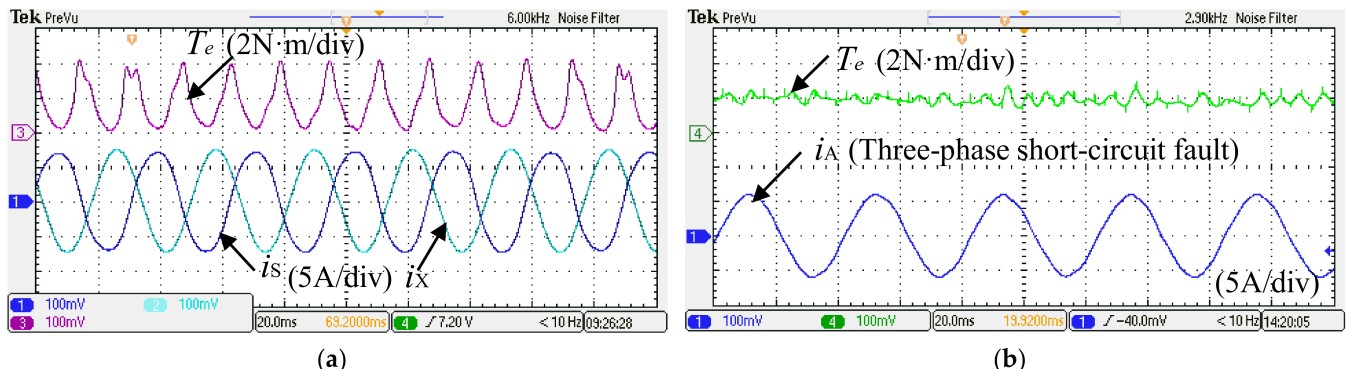

(**a**)               (**b**)

**Figure 19.** Output torque and SCC waveform under single-phase and three-phase short circuit fault. (**a**) Single-phase short circuit fault, (**b**) three-phase short circuit fault.

## 6. Conclusions

This paper systematically analyzes the influence of stator flux barriers on the short-circuit fault tolerance capability of the 12-slot/10-pole dual three-phase PM machine. Compared with conventional machines, the following conclusions can be drawn by theoretical derivation, finite element analysis, and experimental verification.

(1) The appropriate width of the armature tooth flux barriers can increase the PM flux linkage, but the auxiliary tooth flux barriers will reduce the PM flux linkage. Therefore, compared with TSM, the output torque of SL-ATM is increased by 10.3%, while that of SL-ABM is reduced by 10.8%.

(2) Although stator flux barriers can reduce the harmonic leakage inductance of the machine, they can also increase the slot opening leakage inductance and inter-slot leakage inductance. Therefore, the appropriate flux barrier width can improve the self-induction, thus inhibiting SCC and braking torque. Among them, SL-ABM has the smallest SCC and braking torque, followed by SL-ATM.

(3) After entering the fault-tolerant operation mode of three-phase symmetrical short-circuit, in order to keep the machine working at the rated point, the compensation current required by SL-ATM is the minimum, and that of SL-ABM is the maximum.

This paper only studies the influence of the flux barrier width and position on the 12-slot/10-pole dual three-phase PM machine. The influence of the flux barriers on the three-phase or multiphase machines with different pole/slots will be shown in the future research.

**Author Contributions:** Conceptualization, L.L. and K.W.; methodology, L.L.; software, J.L.; validation, L.L. and L.G.; formal analysis, K.W.; investigation, K.W.; resources, K.W.; data curation, J.L.; writing—original draft preparation, L.L.; writing—review and editing, K.W.; visualization, K.W.;

supervision, K.W.; project administration, K.W.; funding acquisition, K.W. All authors have read and agreed to the published version of the manuscript.

**Funding:** This research was funded by the National Natural Science Foundation of China under Project 51977109.

**Data Availability Statement:** Not applicable.

**Conflicts of Interest:** The authors declare no conflict of interest.

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
