# Peer review of "Effect of Flux Barriers on Short-Circuit Current and Braking Torque in Dual Three-Phase PM Machine"

_machines, doi:10.3390/machines10080611_

Round 1
Reviewer 2 Report
Dear Authors,
It was a pleasure to review your paper. Find below some observations regarding it:
1. The aim of the paper is related only to improving the performance of the machine during winding faults. But a machine 99.99% is working in healthy condition and if a winding fault occurs in the shortest time the machine must be replaced. So, most of the working time the investigated short-circuit current and braking torque are not so critical. There are surly negative effects due to the flux barriers on all the significant overall performances of the machine, such as efficiency, maximum power/torque, etc. These should be mandatorily investigated since the reader must know if the toll to be paid for the performance decrease is in balance with the gained advantages.
2. The state-of-the-art survey of the paper must be improved, too. It is based on rather old papers. In this field, dozens of papers are published yearly. The cited papers must be more deeply included within the paper text and a more detailed comparison of their findings should be performed.
3. How do the flux barriers affect the rigidity of the stator and what mechanical measures have to be taken to keep together the stator segments.
4. Section 4.1: the optimization word does not cover what you have been performing since you had not been applying any solid robust optimization method. Your study is more a "try and compare" type of analysis.
5. Your final conclusion ("the rules can be extended to three-phase or multiphase machines with different pole/slot mating.") is not so evident. You better include it in your future studies.
6. You must mandatorily replace RPM with r/min (as the last is the SI unit, and Nm by N·m (since it is a compound unit) – also in the figures!
7. The references are not edited upon the imposed MDPI template!
Reviewer 3 Report
This paper describes the influence of stator flux barriers on the short-circuit current and braking torque of a dual 3-phase permanent magnet synchronous machine. The equivalent model of the flux barriers is created, and the flux barrier width is optimized using finite element analysis. Finally, measurements on a prototype are carried out. This paper is well described except English grammars. English grammars should be checked.
Round 2
Reviewer 1 Report
The authors have made significant changes and addressed most of the reviewer's comments
Author Response
Many thanks to the reviewer for the comments and suggestions on this paper.
Reviewer 2 Report
Dear Authors,
I am still not totally satisfied with the way you answered a part of my questions.
1. I am aware of the importance of fault tolerance of electrical machines, mostly in aerospace applications. But also their efficient work under normal operating conditions should be very important. So if you have former studies on it, please summarise them in the paper or provide a reference.
2. As concerning the stator stability, I am afraid that the designed tails of the stator modules (which are rigidizing the stator segment is only a single point) are not sufficient enough to counterstrike the huge forces blending the modules (see about this peril in DOI: 10.1109/ICEPDS.2018.8571897). Of course, only a mechanical analysis could totally clarify this issue. Did you perform such crucial studies?
